# Early Gastric Cancer: Update on Prevention, Diagnosis and Treatment

**DOI:** 10.3390/ijerph20032149

**Published:** 2023-01-25

**Authors:** Clara Benedetta Conti, Stefano Agnesi, Miki Scaravaglio, Pietro Masseria, Marco Emilio Dinelli, Massimo Oldani, Fabio Uggeri

**Affiliations:** 1Interventional Endoscopy, Foundation IRCCS San Gerardo dei Tintori, 20900 Monza, Italy; 2Department of Surgery and Translational Medicine, Foundation IRCCS San Gerardo dei Tintori, University of Milano-Bicocca, 20900 Monza, Italy; 3General Surgery Unit, Foundation IRCCS San Gerardo dei Tintori, 20900 Monza, Italy

**Keywords:** early gastric cancer, gastric cancer, endoscopic submucosal dissection, *Helicobacter pylori*, upper endoscopy

## Abstract

Gastric cancer (GC) is a relevant public health issue as its incidence and mortality rates are growing worldwide. There are recognized carcinogen agents, such as obesity, tobacco, meat, alcohol consumption and some dietary protective factors. Strategies of early diagnosis through population-based surveillance programs have been demonstrated to be effective in lowering the morbidity and mortality related to GC in some countries. Indeed, the detection of early lesions is very important in order to offer minimally invasive treatments. Endoscopic resection is the gold standard for lesions with a low risk of lymph node metastasis, whereas surgical mini-invasive approaches can be considered in early lesions when endoscopy is not curative. This review outlines the role of lifestyle and prevention strategies for GC, in order to reduce the patients’ risk factors, implement the surveillance of precancerous conditions and, therefore, improve the diagnosis of early lesions. Furthermore, we summarize the available treatments for early gastric cancer.

## 1. Introduction

Gastric cancer (GC) ranks fifth for incidence and fourth for mortality with over 1 million new cases and 768,000 deaths worldwide in 2020. These numbers are predicted to increase by 2040 with 1.77 million new cases and 1.27 million deaths worldwide predicted. The incidence and mortality rates are higher in men (15.8 new cases and 11 deaths per 100,000 person-years) compared to women (7.0 new cases and 4.9 deaths per 100,000 person-years) [1]. Asia reports the highest incidence and mortality rates worldwide, with almost 820,000 new cases and 576,000 deaths in 2020, followed by Europe with more than 136,000 new cases and almost 97,000 deaths in 2020 [1]. Over 90% of GC cases are sporadic, whereas 10% of cases show a familial aggregation and 1–3% arise from inherited cancer syndromes. Adenocarcinoma is the most common type of stomach cancer. Three main histologic subtypes can be described: intestinal type, diffusive type and indeterminate type adenocarcinoma. The WHO classification [2] distinguishes four major types of GC based on the predominant histologic pattern: tubular, papillary, mucinous and poorly cohesive. Moreover, there are some uncommon types of stomach cancer, such as squamous cell carcinoma, non-Hodgkin lymphoma, gastrointestinal stromal tumors (GISTs) and neuroendocrine tumors (NETs).

The prevention of GC is a crucial challenge for public health and the diagnosis of early lesions is essential in order to offer a definitive, mini-invasive treatment and improve survival. 

In the present paper we practically summarize the strategies for the prevention and diagnosis of early gastric cancer (EGC) and the available treatments.

## 2. Prevention

### 2.1. Risk Factors

The main known risk factors for GC are *Helicobacter pylori* (Hp) infection, tobacco smoking, a high intake of meat, obesity and alcohol consumption. Some factors, such as citrus fruit and polyphenol intake, nonsteroidal anti-inflammatory drugs (NSAIDs), statins, metformin and aspirin could have a protective role.

#### 2.1.1. *Helicobacter pylori*

Hp is classified as a class I carcinogen and represents the main environmental risk factor for GC. Up to 89% of non-cardia gastric cancer (NCGC) can be attributed to chronic Hp infection [3]. Indeed, due to its ability to use both the lymphatic and the venous system for dissemination, chronic Hp infection can lead to both gastric adenocarcinoma, by modifying the epithelial-mesenchymal transition, cell migration and cell invasion, and MALT lymphoma, which is often reversible only with Hp eradication [3]. Once infected with Hp and without a proper diagnosis and treatment, the literature reports that the chronic infection will increase the risk of developing GC 1.4 to 4.2 times more than for the general population. Therefore, according to Ford et al., Hp eradication reduces GC incidence (RR = 0.54; 95% CI 0.40–0.72) and mortality (RR = 0.61; 95% CI 0.40–0.92) in healthy individuals [4]. It has been estimated that if all Hp infections were eradicated, approximately 89%, 29% and 74% of NCGC, CGC and gastric non-Hodgkin lymphoma, respectively, would be prevented. The prevalence of Hp is greater in Asia than in Europe (54.7% (51.3–58.1%) 95% CI vs 47.0% (41.8–52.1%); 95% CI) [5]. However, the high incidence of GC in East Asia is not only dependent on the prevalence of Hp infection. In terms of genetic differences, it seems that some Asian populations express point mutations in the sej allele, which promotes bacterial adhesion to the gastric mucosa [6]. Moreover, Asiatic populations appear to have smaller parietal cell masses than western ones, which affect gastric acid secretion and the response to treatment with histamine-2 receptor antagonists and proton pump inhibitors [6]. Nevertheless, no definitive data explains the difference in incidence between Asia and western countries.

Importantly, many countries in Asia have developed screening programs to detect Hp and increase its eradication. Moreover, Japanese and Taiwanese health insurance programs have expanded their coverage for Hp eradication in a subset of patients with chronic gastritis and peptic ulcers, respectively [7]. Table 1 shows the available screening programs for Hp in the world. Unfortunately, no organized screening program for Hp can be found in western countries.

Regarding the long-term effects of Hp eradication, a meta-analysis by Ford et al. analyzed the data from seven randomized controlled trials (RCTs). Among 4206 individuals who received Hp eradication therapy, 1.6% developed GC, compared with 3% of 4117 subjects allocated placebo or no treatment (RR 0.54, 95% CI 0.4–0.72). Thus, they estimated that Hp screening and treatment would result in a gain of over 8.8 million disability-adjusted life-years globally (95% CI 5.7–11.9) [4].

#### 2.1.2. Tobacco

Tobacco smoking is classified as a group 1 carcinogen for GC. In the meta-analysis of Praud et al. the risk of develop GC grows by 25% for current smokers. This data is closely related to the intensity and duration of cigarette smoking; the risk increases by 32% for more than 20 cigarettes/day and by 33% for smoking duration of 40 years or more, as compared to never smokers. The probability of developing GC is similar to that of non-smokers about 10 years after quitting [9]. Interestingly, tobacco smoking seems to be associated the most with cardia GC. Moreover, tobacco smoking seems to be linked to the development of dysplasia, chronic atrophic gastritis and metaplasia. Many studies investigated the genetic basis of the development of GC. Polymorphisms of the GSTT1, SULT1A1, CYP1a1 and NAT2 genes could be involved in an individual’s susceptibility to GC among smokers, such as the hypermethylation of the CDH1 gene which is more expressed in smokers rather than in non-smokers [10]. 

#### 2.1.3. Alcohol

Alcohol drinking is associated with an increased risk of GC, especially for 30 g/day or more, according to heavy-drinker analysis. In the meta-analysis of Deng et al., the risk of GC development in drinkers versus non-drinkers was significantly higher with an odds ratio (OR) of 1.20. An OR of 1.30 was found in the subset of heavy drinkers [11]. Acetaldehyde, the first metabolite of ethanol, could induce DNA lesions by the inhibition of DNA methylation. ALDH2 rs671, a polymorphism of an enzyme involved in alcohol metabolism, seems to increase the concentration of acetaldehyde after drinking. This polymorphism is very common in China, and less so in Caucasian people [12].

#### 2.1.4. Meat

Processed meat (smoked and salted) is often defined as a general carcinogen and the consumption of red meat is associated with the development of non-cardia GC. Indeed, the consumption of red and processed meat is associated with an increased risk of GC, 41% and 57%, respectively [13]. On the contrary, white meat is associated with a low risk of development of GC, with a 20% reported risk reduction. A meta-analysis of dose-response demonstrates a 26% increased risk of GC for 100 g/day of red meat intake and a 72% increased risk for 50 g/day of additional processed meat intake [13]. In the analysis of Ferro et al. based on the Stomach Cancer Pooling (StoP) Project dataset, the higher the consumption of red and processed meat, the greater the risk of GC: OR of 1,85 for 150 g/day of red meat and OR of 1,38 for 50 g/day of processed meat [14]. These results are probably associated with carcinogen compounds such as heme iron and N-nitroso compounds, which help the development of DNA adducts, risks factors for carcinogenesis [14]. Moreover, cooking and storage methods can also increase the risk of GC. Indeed, meat cooked at a high temperature produces heterocycles amine and polycyclic hydrocarbons [15]. Besides, the presence of bacterial plasmids (DNA) from meat were also associated with carcinogenesis through the development of chronic inflammation. On the contrary, as already mentioned, the consumption of white meat was negatively associated with GC, as white meat contains less heme iron and it is a source of polyunsaturated fatty acids (PUFAs) with a lower level of cholesterol than red meat. PUFAs play a role in the prevention of carcinogenesis by apparently reducing chronic inflammation (the inhibition of IL-1 and TNF), blocking COX-2 and inducing apoptosis with antiproliferative action [16]. Interestingly, in animal models, high dietary salt seems to increase chronic inflammation by inducing gastritis. Thus, salted meat could have an increased carcinogenic role in the development of GC [15]. Notably, traditional Asian food tends to be more salted than western food. Furthermore, it could be noticed that in western countries there is greater consumption of white meat. All these factors could play a role in the reported differences in incidence of GC between Asia and other countries.

#### 2.1.5. Obesity

In recent years, obesity has been recognized as a relevant risk factor for the development of many types of cancer, and it will probably overtake the role of smoking in the next few years [17]. In the meta-analysis of Yang et al. a linear association between BMI and the risk of GC is determined [18]. This observation underlines the importance of weight loss at any point of the obesity spectrum, from the overweight to the severely obese. Moreover, obesity is more closely related to cardia GC than non-cardia GC [19]. Indeed, obesity may increase the incidence of gastro-esophageal reflux, which is a recognized cause of Barrett’s esophagus and, eventually, esophageal adenocarcinoma and cardia GC [19]. Besides this, insulin resistance may have a carcinogenic role, but also the role of adipose tissue as an endocrine system has to be stressed. Adiponectin, leptin and IGL1-levels alteration, a different production of glucocorticoids and sex steroids, oxidative stress and a high level of inflammatory markers are all present in obese patients. All these metabolic changes seem to play a role in the risk of cancer. Moreover, all of them could also be linked to a major genetic predisposition [20,21].

### 2.2. Protective Factors

#### 2.2.1. Citrus Fruits and Polyphenol 

Vegetables and fruits have often been associated with a reduced risk of many tumors and also GC. There is suggestive evidence that the protective role of citrus fruits could be associated with bioactive compounds, such as vitamin C, an enzymatic cofactor scavenger of reactive oxygen species and inhibitor of nistrosamine [22,23,24]. The risk of GC seems to be inversely related to citrus fruit consumption. This relationship increases progressively until the consumption of three servings/week. In a large study by the StoP project consortium, the inverse association between citrus fruits and GC is similar for different cancer histotypes. Furthermore, this relationship is stronger in people of low socio-economic status, as a diet rich in citrus fruits could counteract the negative effect of the lifestyle associated with low social classes [25]. Moreover, the potential protective role of citrus fruits could be linked to the presence of flavanones, a class of flavonoids contained only in citrus fruits, such as hespertin and naringenin. There is evidence of their role in vitro; they inhibit the proliferation, migration and invasion of GC cells in a dose and time-dependent manner [26]. Polyphenols are presents in a large variety of vegetables, fruits, cereals, dry legumes and spices. They are divided into five classes. In a large pooled analysis, an inverse association between the high intake of total and specific polyphenols and GC risk was found. Despite the many variables that can create biases in their analysis, polyphenols do seem to have antioxidant and anti-inflammatory effects in vitro. Some of them could even inhibit Hp growth with antimicrobial properties. Lastly, they could trigger the apoptosis of cancer cells [26,27].

#### 2.2.2. NSAIDs, aspirin, statin and metformin

NSAIDs have been studied as potential protective factors against GC. A meta-analysis concluded that NSAIDs, aspirin and non-aspirin NSAIDs could reduce GC risk [28]. The mechanism of protection is unclear and controversial. However, some studies showed that NSAIDs could block the COX-2 pathway, which has an elevated expression in gastric carcinogenesis. Other studies showed that NSAIDs could reduce the risk of developing GC through the interaction of the drugs with B-catenin, WNT signaling, the tumor necrosis factor, polyamine metabolism and the DNA mismatch repair system [29]. Seo et al. analyzed the role of aspirin, metformin and statin in the development of GC: statin and aspirin use were associated with significantly reduced risks of GC development, while metformin was not associated with GC development risk. However, further studies need to be conducted [30]. 

Overall, the role of lifestyle is very important in the prevention of GC. Physicians should invest more time in informing patients about the risks associated with some habits, such as tobacco use and alcohol and meat consumption, and the importance of consuming fruits and vegetables. They should promote the eradication of Hp, and physical activity in order to reduce obesity, and extensively inform the patients about the role of this latter factor in carcinogenesis. General practitioners should be more informed by the specialists about their possible role in the prevention of GC, as they know the patients and their habits the most. This could be achieved by spending more time talking with patients, but also through other initiatives, such as prevention campaigns in schools, meetings between specialists and general practitioners and open days in hospitals for the whole population.

## 3. Screening for Gastric Cancer 

The population benefits worldwide from organized screening programs for many types of cancers. Screening tests constitute a secondary prevention, as they allow for the early detection of cancers even in asymptomatic patients, as well as the detection of early lesions and precancerous conditions. 

Ideally, a screening test for GC would reduce mortality and increase therapeutic success, by allowing the resection of EGC and precancerous lesions. Unfortunately, only a few countries with a high prevalence of GC have organized screening programs for GC. However, considering all the aforementioned risk factors, GC diagnoses are set to increase even in low-prevalence areas. Thus, we believe that knowledge of precancerous conditions is crucial to understand the possible benefits of secondary prevention. Moreover, the availability of serum markers, imaging and, above all, endoscopy, should encourage the clinician to test those patients that have a high risk of GC.

### 3.1. Precancerous Conditions and Lesions

The inflammation–atrophy–metaplasia–dysplasia–carcinoma sequence, known as the Correa cascade, leads to intestinal-type gastric adenocarcinoma [31]. Chronic atrophic gastritis (AG) and intestinal metaplasia (IM) are considered precancerous conditions. Indeed, they represent the background of dysplasia and adenocarcinoma development. The advanced stages of atrophic gastritis should be defined as moderate to marked atrophy or IM in both antral and corpus mucosa. The Operative Link on Gastritis Assessment (OLGA) and Operative Link on Gastritis Assessment, based on the Intestinal Metaplasia (OLGIM) systems were proposed for staging atrophy and IM, respectively. It seems that high versus low OLGIM stages have an increased likelihood of progression to GC, when compared to high versus low OLGA stages [32]. Gastric dysplasia represents the penultimate stage of the gastric carcinogenesis sequence, being a direct neoplastic precancerous lesion. Therefore, during an endoscopy, physicians should search for precancerous conditions and visible lesions to ensure proper surveillance and treatment.

High-definition endoscopy with chromoendoscopy (CE) is better than high-definition white-light endoscopy alone for the diagnosis of gastric precancerous conditions and early neoplastic lesions. After proper training, virtual CE with or without magnification should be used for the diagnosis of gastric precancerous conditions, by guiding biopsies for staging atrophic and metaplastic changes [33].

#### 3.1.1. Atrophic Gastritis 

AG is an asymptomatic, pre-neoplastic condition defined by the slow replacement of the appropriate gastric glandular structures with connective tissue (non-metaplastic atrophy) [31]. The prevalence of AG in western countries is up to 15% [34]. The most common etiologies of AG are chronic infection with Hp and autoimmunity. There are typical endoscopic markers of atrophic gastritis, mainly visible due to the thinning of the gastric mucosa. Firstly, the pale appearance of the gastric mucosa (Figure 1A) and the increased visibility of the vascular architecture, more evident in the antrum, and the loss of gastric folds, more evident in the gastric body. When the endoscopic features of AG gastritis are present, the guidelines recommend performing biopsies for histopathological confirmation and risk stratification [33]. The OLGA protocol suggests a minimum of three biopsies from the antrum/angulus and two biopsies of the corpus. Additional biopsies should be taken from other abnormalities, when present. A surveillance endoscopy every three years should be considered in individuals with advanced atrophic gastritis. Indeed, it is estimated that the risk of the progression of AG to GC ranges from 0.1 % to 0.3 % per year [32]. Moreover, the patients with chronic autoimmune AG are also at an increased risk of type I neuroendocrine tumors [35].

#### 3.1.2. Intestinal Metaplasia

IM is defined as the appearance of intestinal epithelium in the stomach. IM is classified as complete or incomplete. Complete IM is associated with the presence of sucrase, trehalase, leucine aminopeptidase, alkaline phosphatase, goblet cells and Paneth cells, whereas incomplete IM is associated with sucrase, leucine aminopeptidase and goblet cells, but not with trehalase or Paneth cells. Goblet cells in complete IM contain sialomucin, in the small intestine, while those in the incomplete type contain sulphomucin and sialomucin, in the large intestine. IM in the stomach contributes to the physiological mucosal response to injury. Indeed, genes expressing intestine cell-markers are upregulated and the cells at the base of the glands of the body develop a morphology more characteristic of the mucus-producing deep antral glands. The presence of the spasmolytic polypeptide-expressing metaplasia (SPEM) seems to have a key role. SPEM lineages, indeed, have been found at the edges of ulcers in the gastric body [36]. However, when the damage and chronic inflammation continue, as occurs in Hp infection and gastritis, the reparative metaplastic lineages may lead to proliferative pre-neoplastic or dysplastic cells and, eventually, to GC [37]. Hp and gastritis activate the signals of epithelial damage and the release of cytokines. These processes result in inflammatory infiltration into the stomach which eventually leads to the loss of parietal cells. A meta-analysis assessed the association between IM and GC. Overall, 402,636 participants and 4535 GC patients were included. Compared to people without IM, in the group with IM patients were at a higher risk of GC (pooled OR = 3.58, 95% CI 2.71–4.73). GC risk was higher among patients with incomplete IM (pooled OR = 9.48, 95% CI 4.33–20.78). Moreover, IM in the corpus was associated with a higher risk of GC (pooled OR = 7.39, 95% CI 4.94–11.06) than IM in the antrum only (pooled OR = 4.06, 95% CI 2.79–5.91). Thus, patients with IM were at a higher risk of gastric cancer, especially when incomplete and in the corpus [38]. The guidelines suggest endoscopic surveillance for the patients with IM at higher risk of cancer: those with incomplete IM, extensive IM, a family history of GC, ethnic minorities and immigrants from high incidence regions [33]. It is not easy to diagnose IM by endoscopy. In a study involving 150 patients, whitish raised lesions corresponded to IM in 85.7% of cases (*p* < 0.001). This is the most suggestive endoscopic appearance of IM (Figure 1B). The sensitivity and specificity of endoscopy in the diagnosis of IM were 54% and 100%, respectively (*p* = 0.001) [39].

### 3.2. Screening Test to Detect Early Gastric Cancer

Ideally, a screening test should identify high-risk individuals and early lesions. This could help in sparing the use of endoscopy and diagnose EGC at early stages. A sustainable GC screening campaign should address the criteria of reliability and cost-effectiveness. Therefore, particularly in populations with a lower cancer-adjusted mortality rate, such as in western countries, the development of non-invasive tests should be promoted to stratify the risk of GC and to offer screening endoscopy according to the risk category. 

#### 3.2.1. Serum Markers

Biomarkers are non-invasive tests that are objectively measured in different body fluids and evaluated as an indicator of physiological and pathological processes. The conventional markers of GC—such as CEA, Ca19-9, Ca 12-5, Ca 72-4—have been proved to have a low sensitivity and specificity for GC detection and are scarcely useful for the early diagnosis of GC. Therefore, the discovery of novel biomarkers of GC has become a priority in the pipeline of GC research.

One of the most studied biomarkers for the prediction of precancerous gastric lesions has been the serum pepsinogen (sPG). Pepsinogen I (PGI) is secreted by the fundic glands, whereas pepsinogen II (PGII) is produced by fundic, pyloric cells and Brunner’s gland. Both serum PGI and PGII levels increase with the progression of gastritis. As the fundic gland mucosa is reduced due to the gastritis, serum PGI levels gradually decrease, whereas serum PGII levels remain stable. Therefore, the serum PGI/PGII ratio (sPGr) decreases with the progression of gastritis, reflecting the severity of gastric atrophy [40]. The ABC method was one of the earliest non-invasive methods proposed by Fukuda et al. It combines serum immunoglobulin G, anti Hp and serum pepsinogen I (PGI) and pepsinogen II (PGII) levels. The ABC method classifies patients in three different risk categories according to their serological status: (A) IgG anti Hp (−)/PG (−); (B) IgG anti Hp (+)/PG (−); (C) IgG anti Hp (+/−)/PG (+). There is evidence that the ABC classification can correctly stratify patients according to their risk of GC [41]. However, when compared with radiological findings, its discriminative power was shown to be suboptimal. Another biomarker associated with atrophic gastritis is gastrin-17 (G-17), which is secreted by G endocrine cells and depends on gastric acidity. Chapelle et al. [42] validated a panel of stomach-specific biomarkers, including PGI, PGII, G-17 and Hp serology, named GastroPanel, to predict the presence of atrophic gastritis in a population of patients at increased risk of GC. GastroPanel showed a sensitivity of 39.9% and specificity of 93.4%. 

More recently, the interest of the scientific community has been focused on novel molecular biomarkers related to DNA and RNA for early tumor detection. They assess prognosis, monitor tumor burden, predict therapeutic resistance, quantify minimal residual disease and perform real-time cancer management in GC patients [43]. In detail, ’epigenetic’ alterations are the heritable changes in gene expression that do not involve a permanent change in the DNA sequence and they often appear in the early stages of cancer. These alterations, together with genetic events, have emerged as key drivers of cancer development and progression. The most well-studied epigenetic alterations are aberrant DNA methylation, histone modifications and the dysregulated expression of non-coding RNAs. Among them, the dysregulated expression of non-coding RNAs is gaining an important role. They include long non-coding RNAs (lncRNAs) and microRNAs (miRNAs) [43]. 

LncRNAs are RNA transcripts longer than 200 nucleotides with no protein-coding function. They influence tumorigenesis through complex signaling pathways and interact with miRNAs. Their expression levels have been shown to predict the biological behavior of GC cells, including proliferation, invasion, metastasis and drug resistance. A large number of GC-related lncRNAs are reported to be upregulated in GC and have an oncogenic effect by inducing the proliferation and migration of cancer cells by various mechanisms. For example, LINC00152 overexpression stimulates GC cancer cell proliferation, while HOTAIR has a role in tumor immune escape and was shown to regulate HER2 expression by interacting with miRNAs. Again, H19 upregulation is associated with clinical outcomes such as higher invasion depth, advanced TNM stage, regional lymph nodes metastasis and poor prognosis in patients with GC. The role of lncRNAs has also been studied as a possible target for therapy in GC. Indeed, lncRNAs could be targeted through the use of small interfering RNAs for genome editing, and the use of regulatory sequences of lncRNAs in recombinant plasmids could have a therapeutic function. This evidence opens the possibilities of using lncRNAs and their interactions with miRNAs for personalized medicine. The abundance and instability of lncRNAs in the serum limits their wide application in clinical practice so far, but they constitute an interesting and growing research field and a very promising tool for the early detection and therapy of GC [44]. 

MiRNAs are a large group of endogenous small non-coding RNAs of 17–25 nucleotides. Through post transcriptional gene regulation, they influence the modulation of various pathobiological processes in many cancer types, including GC. They bind complementary sequences in 3’-untranslated regions (3’-UTR) of various target messenger RNAs (mRNAs) leading to direct mRNA degradation or translational repression. The stability of miRNAs in the serum makes the liquid biopsy suitable as a diagnostic and prognostic biomarker, as well as a therapeutic target of GC. Different miRNA signatures have been correlated with the development and progression of GC [45]. For example, a 5-miRNA signature (miR-1, miR-20, miR-27a, miR-34 and miR-423-5p) was identified as a diagnostic marker and demonstrated a higher sensitivity than conventional markers (CEA or CA19-9). In a large prospective multicenter study, a 12-miRNA panel for the detection of GC has been developed and has been demonstrated to outweigh the diagnostic accuracy of other non-invasive methods for the diagnosis of GC (AUC 0.84 versus 0.63 of Hp serology, 0.64 of ABC method) [45]. Moreover, miRNA alterations seem to occur early in the pathogenesis of GC, from the earliest phases of the Correa cascade. Thus, they could be useful in identifying patients at high risk of GC development. This observation reinforces their role in the detection of early lesions and thus in the prevention of GC [46]. 

A role of miRNAs in the prediction of resistance in adjuvant chemotherapy for patients with GC has also been explored. In a large cohort of GC patients, miR-1229-3p has been correlated with a higher risk of 5-fluorouracile resistance both in vitro and in vivo. 

Overall, several miRNAs have been associated with the modulation of expression of multidrug resistance-associated proteins leading to a differential response to cytotoxic drugs. These findings suggest that the use of miRNAs might enable the selection of a chemotherapy regimen more tailored to the molecular characteristics of the single patient, as already reported for lncRNAs [47]. 

Overall, up to now, the evidence is still too little to recommend serum markers as a single test for GC screening. Nonetheless, they constitute a fascinating research field and might be useful in combination with endoscopic or radiographic methods to define the target population and the intervals for screening.

#### 3.2.2. Imaging

Japan has conducted photofluorography for GC screening since 1960, leading to a remarkable reduction in mortality due to GC since 1960 of 40–60%. Photofluorography screening is usually performed by the indirect X-ray method with a barium meal, and has showed a good diagnostic performance according to population-based data (sensibility 36.7%; specificity 96.1%) [48]. Abnormal findings might be narrowing, stenosis, deformity, rigidity, barium pooling, irregularity in the gastric area, a change in gastric fold or the presence of polypoid lesions. Nevertheless, all the radiographic abnormalities have to be confirmed and investigated by endoscopy. Korea started its nationwide campaign of GC screening in 2001 in asymptomatic adult patients between 40 and 74 years old, by offering either barium studies or endoscopy, depending on patients’ preferences and comorbidities [49].

#### 3.2.3. Endoscopic Screening

Japan and South Korea are leading countries for the implementation of mass population GC screening. Indeed, the current guidelines from Japan and South Korea recommend endoscopic screening for GC every two years in asymptomatic adults aged 50–75 and 40–75, respectively [50]. The screening interval is based on the doubling time of GC, estimated approximately as 2–3 years [48]. They do not recommend GC screening for adults older than 85 years, as the evidence of benefit is lacking after 75 years [51]. Endoscopy is the gold standard for the diagnosis of GC and EGC and the implementation of technologies such as digital CE has the potential to further increase the detection rate of EGC. The combined use of magnifying endoscopy significantly enhances the diagnostic performance of white-light endoscopy alone (accuracy 96.6% and 64.8%; sensitivity 95.0% and 40.0%; specificity 96.8% and 67.9%, respectively) [52]. Moreover, in a population-based study evaluating the impact of the switch from a radiographic to an endoscopic-based mass screening for GC, a significant reduction in GC-related death rates was found (5.0/year versus 2.1/year). Accordingly, in the last decade in Asian countries there has been an increase in the use of endoscopy over imaging screening, even if the updated Japanese and Korean guidelines do not strongly suggest endoscopy over radiology [53]. The main aspects limiting the wide spread use of endoscopy screening are the risk of over-diagnosis, the potential occurrence of complications and the availability of skilled endoscopists and adequate endoscopes. As preliminary data suggest, the quality of routine endoscopy could have a differential effect on the prevalence of GC, due to more careful evaluation and the better detection of pre-cancerous lesions. Indeed, up to 10% of missed precancerous lesions of the stomach have been reported in the three years before GC diagnosis. This has highlighted the need for standard quality measures for upper gastrointestinal endoscopy, such as the delivery of proper pre-procedure fasting instructions, the definition of a minimum duration of endoscopy and adequate photo documentation of the anatomical landmarks and lesions limiting blind areas (Table 2) [54].

## 4. Treatment of Early Gastric Cancer

### 4.1. Endoscopy

#### 4.1.1. Indications for Endoscopic Resection of Early Gastric Cancer

Generally, the indication for an endoscopic resection (ER) is determined by the risk of lymph node metastasis (LNM) and the possibility of achieving an en bloc resection. Basically, the recommended endoscopic treatments for EGC lesions are endoscopic mucosal resection (EMR) and endoscopic submucosal dissection (ESD). The factors that guide the physician in choosing the most suitable endoscopic treatment are the histopathological type, the size of the lesion, the depth of invasion and the presence of ulceration. Overall, intramucosal adenocarcinomas (pT1a) have a 2–5% incidence of LNM, whereas a submucosally invasive adenocarcinoma (pT1b) has a risk of 10–25%. However, if certain histological characteristics are met, the risk of LNM is minimal. Table 3 summarizes the indications for EMR and ESD of EGC [55]. 

Moreover, based on the risk of LNM and the long-term outcomes of an ER compared to a surgical resection, the indications can be distinguished as absolute, expanded (only in cases of differentiated-type carcinomas) and relative [56]. For intramucosal undifferentiated-type adenocarcinomas, an ER can be considered for a lesion < 2 cm, without ulcers. However, the choice should be individualized after considering the surgical risks and patient conditions (Table 4). 

Moreover, the resection of a > 30 mm gastric adenocarcinoma with superficial submucosal invasion (SM1) or with ulceration should be considered at high risk of non-curative resection. Thus, there should be complete staging and strong consideration for surgery [57,58]. 

Table 5 summarizes the criteria for curativeness of ER and its risk stratification.

#### 4.1.2. Endoscopic Preoperative Diagnosis

The evaluation of EGC can be very difficult. Therefore, narrow-band imaging and CE, (vital or digital), are two methods that magnify the endoscopy and improve accuracy in the evaluation of the horizontal extent of EGC [59]. The Japanese Gastroenterological Endoscopy Society proposed a magnified endoscopy simple diagnostic algorithm (MESDA) for GC [60] based on the study of the microvascular pattern (MV) and the microsurface (MS) pattern. The MV pattern analyzes the subepithelial capillary, collecting venule and pathological microvessels, whereas the MS pattern is identified by the marginal crypt epithelium, crypt opening and an intervening part between crypts. Therefore, the detection of an abrupt change in MV and/or MS patterns in the mucosal surface, should define the limit of the lesion. MESDA has a sensitivity, specificity, diagnostic accuracy, positive predictive value and negative predictive value of 83%, 89%, 95%, 79% and 99%, respectively [61]. Nevertheless, a clear demarcation line between the lesion and the surrounding mucosa can be difficult to find, especially in undifferentiated intramucosal cancer. Therefore, in those cases, biopsies from the mucosa surrounding the lesion are also recommended. 

The diagnosis of the depth of EGC lesions is even more difficult and essential for the pre-operative diagnosis, as the presence of an SM invasion of more than 500 µm defines the T1b2 (SM2 invasion). Size > 30 mm, the presence of uneven surface, margin elevation and mucosal fold convergence seem to be useful in the depth diagnosis of EGC during a conventional endoscopy. However, even if the accuracy of depth diagnosis is reported to be between 82 and 97%, these criteria largely depend on the endoscopist’s experience [62]. 

Endosonography (EUS) is nowadays largely available and its use has been proposed to assess the depth of the invasion. Up to now, the use of EUS does not seem superior to conventional endoscopy for a depth diagnosis of EGC. However, its use can be considered in combination with a conventional endoscopy (Figure 2). A multicenter prospective study of 175 patients evaluated the additional value of EUS for EGC suspected of SM invasion. In each case, the diagnosis was first made using a conventional endoscopy followed by EUS, and finally confirmed using a combination algorithm. The accuracy rates of conventional endoscopy, EUS and a combination of the two in 108 differentiated-type lesions were 51.9%, 77.4% (*p* < 0.001) and 79.6% (*p* < 0.001), respectively. Therefore, EUS seems to significantly improve the accuracy of conventional endoscopy in SM2 low-confidence lesions, but not in SM1 lesions or in SM2 high-confidence lesions, as an irregular surface, submucosal tumor-like elevation and non-extension signs were found as significant independent markers of pSM2 [63]. Another promising improvement in the diagnosis and assessment of EGC could be the use of artificial intelligence. However, it is still in the earliest stages of development. 

#### 4.1.3. Endoscopic Mucosal Resection and Submucosal Dissection

After considering the patient’s condition, the features of the lesion and the experience of the endoscopist, if ER is indicated, EMR or ESD should be chosen. 

EMR was the first endoscopic method used for the ER of EGC. Basically, it is a resection method where the lesion is elevated with a lifting agent. Then the lesion is resected with a metal snare using high-frequency diathermy. Even if the technique has been improved with high performing snares, the cap-assisted method (Figure 3) and under water EMR, it has been reported that en bloc resection rates for lesions >10 mm are significantly lower for EMR than for ESD [64]. 

Indeed, ESGE recommends ESD as the first option, mainly to provide an en bloc resection with accurate pathology staging and to avoid missing important histological features [65]. ESD was proven to have a significantly higher rate of en bloc resection (>90%), a lower recurrence rate (< 5%) and similar post-procedural bleeding. However, the perforation risk and the procedural duration are the main disadvantages of ESD, especially in western countries. Nevertheless, meta-analyses showed that ESD should also be considered in the treatment of gastric superficial neoplastic lesions, i.e. dysplastic lesions of any size, as it allows high rates of en bloc R0 curative resection with a good level of safety [66].

ESD is a method that follows precise steps. The traditional technique is firstly, to mark the normal mucosa surrounding the EGC to increase the visibility of the margins of the lesion, then the lesion is elevated with a long-lasting lifting agent. Thus, the mucosa surrounding the tumor is pre-cut (Figure 4A). The lifting protects the muscle layer from mechanical and thermal perforation during the incision of the normal mucosa close to the lesion. Then the submucosal layer beneath the lesion can be dissected, with the help of a distal cap (Figure 4B,C). 

The aim is to achieve the en bloc resection of the lesion. The main ESD techniques described are traditional ESD, with or without traction techniques, the pocket creation method and tunnelling. A systematic review and meta-analysis examined the complete resection process, length of hospital stay (LOHS), adverse events (AEs), serious AEs, recurrence, five-year overall survival (OS) and five-year cancer-specific survival (CSS) in patients with EGC. ER was associated with a lower incidence of AEs (RD = −0.07, 95% CI = −0.1, −0.04, *p* < 0.0001) and shorter LOHS (95% CI −5.89, −5.32; *p* < 0.00001) than surgery (n = 12,850). However, ER was associated with lower complete resection rates (RD = −0.1, 95% CI = −0.15, −0.06; *p* < 0.00001) and higher rates of recurrence (RD = 0.07, 95% CI = 0.06; *p* < 0.00001). Importantly, no significant differences were found between surgery and ER in terms of five-year OS (RD = −0.01, 95% CI = −0.04, 0.02; *p* = 0.38), five-year CSS (RD = 0.01, 95% CI = 0.00, 0.02; *p* < 0.17), or the incidence of serious AEs (RD = −0.03, 95% CI = −0.08, 0.01; *p* = 0.13) [67]. Importantly, patients with lesions with positive lateral margins, in the absence of positive vertical margins or submucosal or lymphovascular invasion, can be managed with further ER, given the very low risk of LNM [55]. 

### 4.2. Surgery

When ER is not feasible and the lesion does not meet the ER criteria, patients with ECG need to be directed to surgery. New minimally-invasive approaches and personalized treatment options potentially offer a favorable long-term oncological outcome and a good QOL for patients with EGC.

#### 4.2.1. Minimally invasive Approaches

GC at an early stage has an excellent prognosis when treated with surgery, with a five-year survival rate of over 90%. Therefore, recent studies have focused on postoperative complications and recovery after surgery. A recent meta-analysis by Wei Lu et al. [68] compared the long-term clinical outcomes of laparoscopy distal gastrectomy (LADG) versus open distal gastrectomy (ODG) for EGC. They found no differences in oncological outcomes between groups. The mortality and relapse rate were similar between the LADG and ODG group. The LADG group had significantly fewer short-term complications (relative risk (RR) = 0.57; 95% CI, 0.44–0.76) and a lower RR of long-term complications (RR = 0.63, 95% CI: 0.39–1.01) compared to ODG. Moreover, LADG also improved outcomes by reducing blood loss and wound length and accelerating post-operative recovery, leading to a shorter postoperative hospital stay. Therefore, laparoscopic gastrectomy showed better short-term outcomes than conventional open gastrectomy for EGC [68]. 

New minimally invasive techniques such as robotic-assisted laparoscopy may overcome some of the intrinsic limitations of a traditional laparoscopy. Robotic technology has the advantage of improved dexterity, stability (tremor filter) and stereoscopic visualization. This is useful when a precise dissection is needed, such as during a lymphadenectomy in gastric surgery. Recent studies have reported that the median number of harvested lymph nodes during a D2 lymphadenectomy is similar to that of an ODG and, in some cases, superior to that of a LADG. However, robotic-assisted laparoscopy has not yet shown advantages for EGC. Indeed, a higher number of harvested lymph nodes does not necessarily improve the overall survival, even if it improves the staging accuracy [69].

#### 4.2.2. Function-preserving Gastrectomy

Despite the aforementioned minimally invasive approaches, post gastrectomy syndrome often occurs, decreasing the QOL of patients. To overcome this issue, patients with EGC could benefit from function-preserving gastrectomies. In particular, proximal gastrectomy (PG) and pylorus-preserving gastrectomy (PPG) could be offered to ECG patients instead of total gastrectomy (TG) and distal gastrectomy (DG), respectively. 

A meta-analysis by Xu et al. [70] showed that PG was superior to TG in terms of operative time, intraoperative blood loss and postoperative weight loss. The incidence of complications after PG, such as pancreatic fistulas, bleeding and anastomotic leakage is similar to that after TG (odds ratio 0,56 95% CI: 0.19–1.6; 1.13 95% CI: 0.32–3.9; 0.72 95% CI 0.42–1.26, respectively). Reflux esophagitis, however, seems to be higher after PG than after TG. For this reason, fundoplication after PG can be considered as an option. In PPG, the length of the pyloric cuff retained usually measures between 3 and 4 cm and a gastro-gastro anastomosis is the most used technique for reconstruction.

A meta-analysis by Mao et al. [71] showed that PPG has several advantages over DG such as a lower incidence of dumping syndrome, bile reflux, anastomotic leakage, gastritis and weight loss. The five-year overall survival rate of PG and PPG was reported to range from 94% to 97% and from 96.3% to 98%, respectively.

#### 4.2.3. Sentinel Node navigation Surgery

The presence of lymph node metastasis is an important prognostic factor of EGC and its incidence is about 10% overall. Recently, some studies have evaluated the long-term prognosis of ECG patients undergoing a function-preserving gastrectomy combined with a sentinel node (SN) biopsy. 

The SN is the first possible node of metastasis, as it is the first lymph node to drain the lymphatic flow from the primary lesion. When the results of the histological examination of the SN are negative, all the regional lymph nodes can be predicted to be negative for metastasis. Therefore, an unnecessary radical lymph node dissection can be avoided by performing a SN biopsy combined with surgical treatment for patients with EGC. SN mapping can be performed by using both radioactive colloids and blue or green dye injected via endoscopy into the submucosal layer surrounding the primary lesion. A meta-analysis by Wang et al. [72] covering 38 studies with over 2100 patients involved reported that the detection rate of the SN was 94% and the accuracy of SN diagnosis was 92%. Nevertheless, unlike with breast cancer, after initial gastrectomy there is no room for additional nodal dissection. Moreover, when facing GC, intraoperative diagnostic methods for lymph node micro-metastasis are lacking. For this reason, up to now, a lymphadenectomy guided by SN biopsy should not be omitted.

To overcome these issues, lymphatic-basin dissection has been proposed. The lymphatic basin is identified by dye mapping, removed en bloc and then sent to the pathologist for intraoperative diagnosis. If the lymph nodes are positive, D2 lymphadenectomy is performed, whereas with negative lymph nodes additional lymph node dissection is avoided. EGC surgery treatment will eventually shift from gastrectomy with D1+ lymphadenectomy to a minimally invasive, tailored, function-preserving gastrectomy, in which a lymph node diagnosis can be made intraoperatively.

### 4.3. Comparison between Endoscopic Resection and Surgery

Several studies have compared ER and surgical treatment for EGC. A study by Qian et al. examined the clinical outcomes of ESD and surgical resection for EGC in China [73]. The indications for ESD in the study were all differentiated intramucosal carcinomas without ulceration, differentiated intramucosal cancers 3 cm in diameter with ulceration, undifferentiated intramucosal cancers 2 cm in diameter without ulceration and superficial submucosal invasive cancer (sm1) of differentiated type without ulceration and tumor diameter of 3 cm. The results showed that the patients who underwent ESD had a shorter hospital stay, shorter operative time and fewer adverse events, such as bleeding, gastropathy and perforation (*p* = 0.001). Furthermore, no statistically significant differences were found between the two groups (surgery and ESD) in terms of five-year overall survival and five-year recurrence-free survival: 96.1% versus 91.4% (*p* = 0.08) and 95.8% versus 91.4%, for the surgery and ESD group, respectively. A recent meta-analysis [74] considered nine studies. ESD was performed following the absolute and expanded indications. ER was associated with a shorter hospital stay and fewer postoperative complications compared to gastrectomy. The recurrence rate was higher for ER than for surgery (hazard ratio (HR) = 3.56 95% CI 1.86–6.84) due to the fact that metachronous GC developed only in the patients who had undergone ER treatment. However, the patients who developed metachronous GC could be treated with ER again. The second treatment did not affect the overall survival of the patients. Indeed, overall survival was similar in the ESD and surgery group (HR = 0.84 95% CI 0.63–1.13). A study by Tae et al. [75] compared the subjective quality of life after ESD and surgery for EGC. The inclusion criteria for the ESD group were an en bloc resection with free vertical and lateral margins and a tumor confined to the submucosal layer without lymphovascular invasion and lymph node metastasis. They examined the QoL of EGC survivors over a five-year period after ESD or laparoscopic subtotal gastrectomy, without recurrence. The QoL related to the gastric cancer subscale (GCS) was significantly higher in patients who underwent ESD than in those who underwent surgery (*p* = 0.001). Therefore, as already discussed, ER should be considered as the first-line treatment for EGC, when the indications for ER are met.

## 5. Conclusions

GC is a serious public health issue, as its mortality and morbidity rates remain very high and are continuously growing worldwide. Prevention is the key to reducing mortality. Firstly, lifestyle plays an important role in carcinogenesis and the physician should focus more on its modification, with the reduction of risk factors. Furthermore, the eradication of Hp should be encouraged, when possible. Moreover, in patients with a higher risk of GC the use of available screening tests, such as serum markers or endoscopy, could be considered. The surveillance of precancerous conditions and a careful endoscopic examination could improve the prevention and detection of GC at early stages. Indeed, the detection of early lesions is crucial, as the treatment of EGC is minimally invasive. ER is nowadays the gold standard for EGC with a low risk of LNM and surgical minimally invasive approaches offer a favorable long-term oncological outcome and a good QoL for patients with EGC. Nevertheless, the main obstacle to detecting early lesions is the lack of systematic screening tests for GC worldwide, even if a large knowledge base of precancerous conditions is nowadays available. Therefore, wide population-based screening for precancerous conditions and lesions should be implemented and encouraged.

## Figures and Tables

**Figure 1 ijerph-20-02149-f001:**
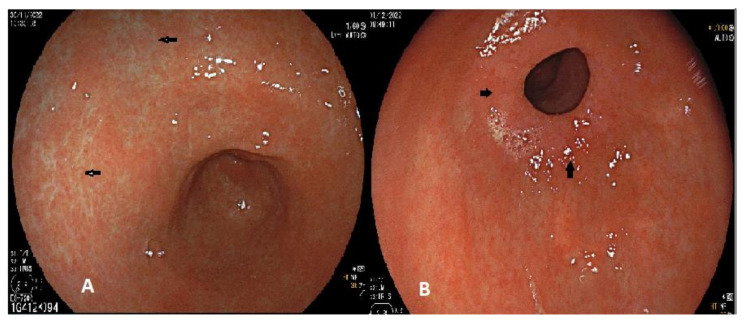
Gastric antrum with atrophic mucosa (**A**) and with IM, visible as whitish lesions surrounding the pylorus (**B**). Arrows indicate the areas with more atrophy (**A**) and the areas with IM (**B**), respectively.

**Figure 2 ijerph-20-02149-f002:**
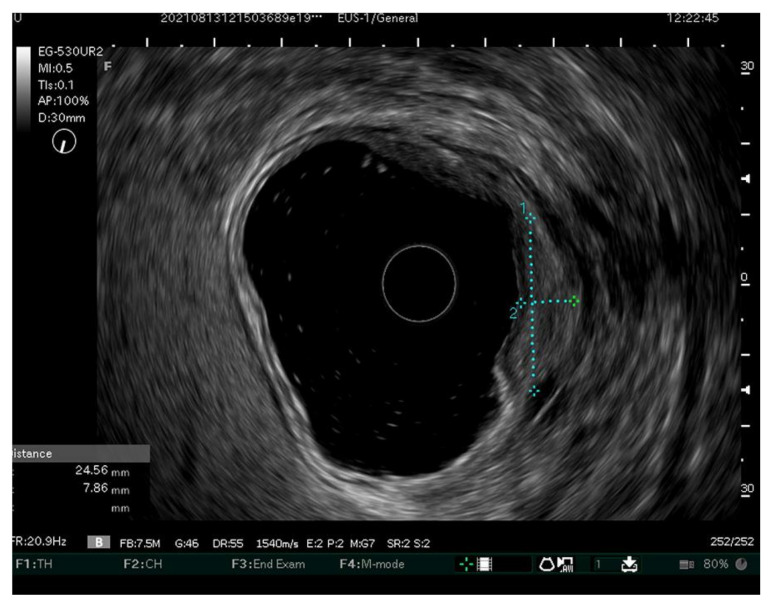
EUS image of a gastric lesion 24.5 × 7.8 mm in size. The endoscopy result was low-confident for SM invasion. EUS was suggestive of SM1 invasion only. After ESD, SM1 invasion was confirmed.

**Figure 3 ijerph-20-02149-f003:**
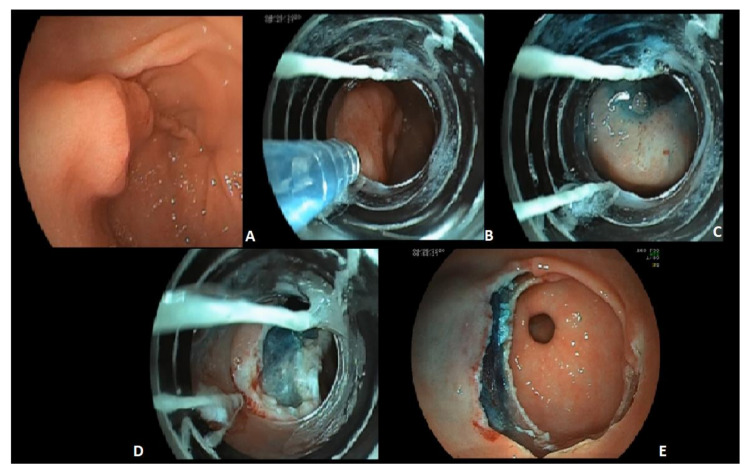
Cap-assisted EMR of a T1a early gastric cancer 1 cm in size, located in the antrum. The figures show: the entire lesion (**A**); the endoscopic device for cap-assisted EMR and the needle filled with the lifting solution approaching the lesion (**B**); the lesion after lifting, before the suction in the cap and the resection (**C**); the antrum immediately after the resection (**D**); the surface of the antrum after the resection and the exposed submucosal layer (**E**).

**Figure 4 ijerph-20-02149-f004:**
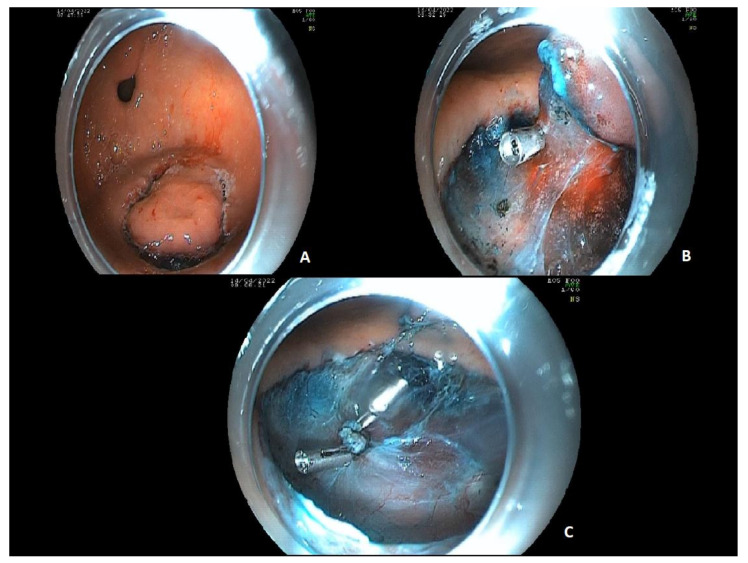
Traditional ESD of a high-dysplasia lesion, 20 mm in size, located in the gastric antrum. In (**A)** the mucosal layer surrounding the lesion is already pre-cut, after the injection of a lifting agent, which can be seen in blue. In (**B**) most of the lesion is already dissected. A clip had been positioned to stop arterial bleeding which occurred during the dissection procedure. The lesion can be seen in the upper part of the figure, almost all dissected. In the lower part of the figure the submucosal layer can be seen. In (**C**) the dissection is completed. The clips were positioned during the ESD to stop an arterial bleed. The submucosal layer is intact after the dissection.

**Table 1 ijerph-20-02149-t001:** Available screening programs for Hp worldwide.

Country	Screening Age	Beginning of Screening	Screening Interval	Strategy	Expected or Demonstrated Benefits
Japan	20 years	2013	Once	Hp infection diagnosed at endoscopic screening	6% reduction in GC mortality in 2016 [8].
Republic of Korea	40–65 years	2014	Once	Urea breath test (UBT) screening	To reduce the incidence of GC through Hp eradication.
China	18 years	2022	Once	Through UBT screening for parents; reach children for Hp testing.	To prevent Hp spread among family members and thus reduce GC incidence and related costs.
Taiwan	30 years	2004	Every 2 years	UBT screening	53% reduction in GC incidence and 25% reduction in GC mortality [7].

**Table 2 ijerph-20-02149-t002:** Standard performance measures for upper gastrointestinal endoscopy. Adapted from Bisschops et al, Endoscopy 2016 [54].

Key Performance Measures	Minor Performance Measures
Fasting instructions prior to endoscopy	Minimum 7-minute procedure time for first diagnostic endoscopy and follow-up of gastric intestinal metaplasia
Documentation of procedure duration	Minimum 1-minute inspection time per cm circumferential Barrett’s epithelium
Accurate photo-documentation of anatomical landmarks and abnormal findings	Use of Lugol chromoendoscopy in high-risk patients to exclude a second primary esophageal cancer
Accurate application of standardized disease-related terminology	Application of validated biopsy protocol to detect gastric intestinal metaplasia (MAPS guidelines)
Application of Seattle protocol in Barrett’s surveillance	Prospective registration of Barrett’s patients
Accurate registration of complications after therapeutic endoscopy	
MAPS (management of patients with precancerous conditions and lesions of the stomach)	

**Table 3 ijerph-20-02149-t003:** Adapted from the guidelines for EMR and ESD of EGC [55].

Depth of Invasion	Ulceration	Differentiated Type		Undifferentiated Type	
cT1a(M)	UL0	≤20 mm absolute indications for EMR/ESD	>20 mm absolute indications for ESD	≤20 mm absolute indications for ESD	>20 mm relative indications
	UL1	≤30 mm absolute indications for ESD	>30 mm relative indications	Relative indications	
cT1b(SM)		Relative indications		Relative indications	

EMR: endoscopic mucosal resection, ESD: endoscopic submucosal dissection, cT1a (M): intramucosal cancer (preoperative diagnosis), cT1b (SM): submucosal invasive cancer (preoperative diagnosis), UL: finding of ulceration (or ulcer scar), UL0: absence of ulceration or ulcer scar, UL1: presence of ulceration or ulcer scar [55].

**Table 4 ijerph-20-02149-t004:** Absolute, expanded and relative indications for endoscopic resection of EGC [55].

Criteria	Absolute Indication	Expanded Indication	Relative Indication
Risk of lymph node metastasis	Less than 1%	Less than 1%	Other than absolute/ extended
Long-term outcomes compared to surgical resection	Equal	Equal, poor evidence	Surgery cannot be recommended Accurate histopathological diagnosis cannot be established due to the patient’s general condition cannot be established due to the patient’s general condition

**Table 5 ijerph-20-02149-t005:** Criteria for curativeness of ER [56].

	Criteria		
Risk of ER	En Bloc Resection R0	Lymphovascular Invasion	Pathology and Size
Curative/very low (LNM risk < 0.5 %–1 %)	Yes	No	Dysplastic/pT1a, differentiated; Any size if no ulcers and ≤ 3 cm if ulcers.
Curative/low risk (LNM risk < 3 %)	Yes	No	pT1a, poorly differentiated or undifferentiated, size ≤ 2 cm, no ulcers; pT1b, SM invasion ≤ 500 µm, differentiated, size ≤ 3 cm.
Local-risk resection (very low risk of LNM but increased risk of persistence/recurrence)	No/positive lateral margin of a lesion otherwise meeting very low risk criteria; No/positive lateral margin without SM invasion.	No	pT1b, SM invasion ≤ 500 µm, well-differentiated; Size ≤ 3 cm.
Noncurative/high risk Any lesion with any of the criteria	Positive vertical margin	Yes	Deep SM invasion (>500 µm); Ulceration or size > 2 cm, in poorly differentiated lesions; Size > 3 cm in pT1b differentiated lesions with SM invasion < 500 µm, or intramucosal ulcerative lesions.

## Data Availability

No new data were created.

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
