# Peer review of "Early Gastric Cancer: Update on Prevention, Diagnosis and Treatment"

_ijerph, 2023, doi:10.3390/ijerph20032149_

Round 1

Reviewer 1 Report

The article entitled "Early gastric cancer: update on prevention, diagnosis and treatment" is very interesting, however, the authors should improve the review and show a molecular update in gastric cancer.  The article should show molecular proposals and miRNAs used as biomarkers in gastric cancer as well as aberrant expression of microRNAs (miRNAs) is the hallmark of gastric cancer. Non-coding RNA in gastric cancer and its involvement in drug resistance should described in this review.

Author Response

Thank you for your comments. 

Reviewer 2 Report

Although the purpose of the review is good and timely important, authors are asked to provide a well-written summary in the abstract.
It lacks substance and conveys no message to readers in general.
It is entirely descriptive.
The provided figures are poorly explained.
If possible, do share additional figures.
Describe and list the comparison study's findings. .

Author Response

Thank you for the comments.

Reviewer 3 Report

1. This review study on early detection, prevention, and treatment of Gastric ulcer is scientifically sound.

2. In my opinion the important risk factors e.g. Chronic atrophic gastritis and Gastric intestinal metaplasia should be described in a bit of detail with separate headings.

3. Each Figure  should be described as a separate Legend

4. Include the list of Abbreviations in manuscript

Author Response

Thank you for the comments.

Reviewer 4 Report

This is a well-written manuscript that provides important information about early gastric cancer. The writing is objective and summarized the main issues to be discussed.

Considering Third Expert Report on Diet, Nutrition, Physical Activity,
and Cancer ( from The World Cancer Research Fund/American Institute for Cancer Research, 2020), I recommend that the section about diet be more elaborate, focusing not only on meat but on processed meat consumption as well as salt-preserved meat consumption. Fruit consumption is reported as a protective factor and can also be better developed.

In addition, obesity is now considered to be one of the main risk factors, even surpassing smoking in the next few years.

I believe that review articles, speacially in cancer field, needs to focus on modifiable risk factors, as well as describe others like biomarkers, screening and treatment.

Author Response

Thank you for your comments.

Round 2

Reviewer 1 Report

The authors should review the bibliography to include more molecules important that participate in gastric cancer (MiRNA as potential biomarkers and therapeutic targets for gastric cancer World J Gastroenterol 2014; 20(30): 10432-10439 [PMID: 25132759 DOI: 10.3748/wjg.v20.i30.10432] ) and include to lncRNAs as potential biomarkers and therapeutic targets for gastric cancer

Author Response

Thanks for your comments.

Reviewer 2 Report

reduce the number of references to maximum75 and organize the image in few complex figure.

Author Response

Thanks for your comments.
